# A user-friendly herbicide derived from photo-responsive supramolecular vesicles

Cheng Gao[1], Qiaoxian Huang[1], Qingping Lan[1], Yu Feng[1], Fan Tang[1], Maggie P.M. Hoi[1], Jianxiang Zhang[2], Simon M.Y. Lee[1] & Ruibing Wang [1]

Paraquat, as one of the most widely used herbicides globally, is highly toxic to humans, and chronic exposure and acute ingestion leads to high morbidity and mortality rates. Here, we report user-friendly, photo-responsive paraquat-loaded supramolecular vesicles, prepared via one-pot self-assembly of amphiphilic, ternary host-guest complexes between cucurbit[8]uril, paraquat, and an azobenzene derivative. In this vesicle formulation, paraquat is only released upon UV or sunlight irradiation that converts the azobenzene derivative from its *trans*- to its *cis*- form, which in turn dissociates the ternary host-guest complexations and the vesicles. The cytotoxicity evaluation of this vesicle formulation of paraquat on in vitro cell models, in vivo zebrafish models, and mouse models demonstrates an enhanced safety profile. Additionally, the PQ-loaded vesicles' herbicidal activity against a model of invasive weed is nearly identical to that of free paraquat under natural sunlight. This study provides a safe yet effective herbicide formulation.

---

[1] State Key Laboratory of Quality Research in Chinese Medicine, Institute of Chinese Medical Sciences, University of Macau, Macau 999078, China. [2] Department of Pharmaceutics, College of Pharmacy, Third Military Medical University, Chongqing 400038, China. Correspondence and requests for materials should be addressed to S.M.Y.L. (email: simonlee@umac.mo) or to R.W. (email: rwang@umac.mo)

Paraquat (PQ) is one of the most widely used herbicides globally in agriculture and gardening. Its oxygen-activated effects take place quickly upon contact with green tissue and are non-selective[1]. In photosystem I (PSI), plastocyanin transfers its electron to ferrodoxin and finally to nicotinamide adenine dinucleotide phosphate (NADP$^+$). PQ ion binds near ferrodoxin site in the PSI and accepts an electron to form PQ monocation free radical (PQ$\cdot^+$), which can stop electron transport to NADP$^+$ and inhibit normal functioning of PSI. As a result, PQ$\cdot^+$ initiates a series of reactions leading to cell membrane disruption and plant death[2,3]. Once PQ comes into contact with the soil, it is deactivated and immobilized[4]. Thus, PQ is an ideal and unsubstituted herbicide in terms of the weed control efficacy.

However, the toxicity of PQ to humans is becoming an increasingly serious problem[5–7]. One small sip can be fatal and currently there is no effective antidote clinically available. The World Health Organization (WHO) has classified PQ as a class II toxic substance (moderately hazardous)[8]. Worldwide, PQ accounts for 20 deaths per million persons[9]. In Korea, PQ has caused an estimated 2000 toxic ingestions annually, via oral, dermal, and inhalation routes[9], with an associated 60–70% mortality rate during the past three decades[9].

As PQ is a rapidly-acting, non-selective, inexpensive herbicide, in spite of being banned or restricted in several developed countries, it is still widely used in many other developed nations and much of the developing world nowadays. During the past several decades, many efforts have been devoted to developing user-safe PQ formulations that are ideally, equally herbicidal. For instance, Syngenta in Sri Lanka designed a PQ formulation, named Inteon®[10], in 2004, with the addition of three components, including alginate, emetic agent, and a dye, into the aqueous solution of PQ. Alginate was used to thicken the formulation in the gastric environment of the stomach for reducing gastric absorption, the emetic agent was used to stimulate the vomitus and purgative reactions for elimination of PQ from the stomach or intestines, if ingested orally, and the dye was used to distinguish the herbicide from other consumer products to avoid misuse. This product has been commercially successful, however, this formulation has exhibited diminished weed control performance in comparison with that of the PQ and with accidental or suicidal ingestions, the survival rate was only moderately improved[11]. Additionally, this PQ formulation does not eliminate toxic effects of PQ with chronic exposure or suicidal ingestion by other administration routes. Huang et al. reported that water-soluble pillar[6]arene was able to reduce the toxicity of PQ in vitro. However, the toxicity was not evaluated in vivo and there was no demonstration of the herbicidal efficacy of this formulation[12]. Dinis-Oliveira and co-workers developed a formulation of PQ with lysine acetylsalicylate (LAS), which demonstrated low mammalian toxicity and effective herbicidal activity[13]. The survival rate of Wistar male rats orally administered with 125 mg kg$^{-1}$ of PQ increased from 40 to 100% after mixing PQ with 316 mg kg$^{-1}$ of LAS (PQ:LAS molar ratio of 1:2)[13]. However, the safety evaluation in this study was limited to survival rate of rats only, without any detailed toxicity evaluations at molecular level (e.g., inflammatory factors, and renal and hepatic function biomarkers) or organ levels (e.g., histological examinations) in vivo. Therefore, a user-friendly PQ formulation that is nontoxic or much less toxic than PQ with strong evidence from systemic safety evaluations in vivo, but with comparable herbicidal effects with that of PQ, is still highly desirable.

On the other hand, many supramolecular vesicles based on cyclodextrins[14], pillararenes[15] and cucurbit[n]urils[16–19] were studied in recent years, for their potential applications in drug or herbicide delivery due to their environmental responsiveness[20]. For instance, Wang and co-workers developed pH-responsive supramolecular vesicles based on the water-soluble pillar[6]arene for controlled drug delivery[21]. Huang and co-workers constructed a dual-responsive supra-amphiphilic polypseudorotaxanes from pillar[7]arene for the controlled release of calcein molecules[22]. Scherman and co-workers reported orthogonal stimuli-responsive single supramolecular systems based upon cucurbit[8]uril-mediated heteroternary complexation with viologen and azobenzene derivatives as functional guests in water[23]. The complexation process was reversed by ultraviolet (UV) light-induced isomerization of azobenzene, the principle of which has been extensively utilized in many other photo-responsive molecular or nano-systems[14,24]. This has inspired us to design a sunlight (UV)-responsive vesicle formulation of PQ that may only release the herbicide under sunlight (UV) with well-preserved weed control efficacy, whereas ingestion of the formulation would be much less toxic to humans.

Herein, we report a photo-responsive, PQ-loaded vesicle that is prepared via our facile one-pot approach by the self-assembly of amphiphilic ternary host-guest complexes between cucurbit[8]uril (CB[8]), PQ, and an azobenzene derivative (trans-G). This user-friendly herbicide, under natural sunlight, imposes its herbicidal effects that are comparable to that of free PQ, whereas ingestion of this formulation by both zebrafish and mouse models affords much safer toxicity profiles than those of free PQ. This study provides a safe yet effective photo-responsive supramolecular vesicle formulation of PQ.

## Results

**Ternary complexation in water.** It is well known that PQ, CB[8] and azobenzene form 1:1:1 ternary complexes[19,24,25]. In order to obtain an amphiphilic supramolecular complex, a hydrophobic azobenzene derivative, trans-G (Fig. 1a), was firstly prepared (the synthetic method is described in the Methods section). The ternary complexation between trans-G and PQ ⊂ CB[8] in water was confirmed by Job plot method under UV-vis spectroscopy. As shown in Supplementary Fig. 1, the binding ratio between trans-G and PQ ⊂ CB[8] was 1:1, consistent with the previous reports[23]. Isothermal titration calorimetry (ITC) studies were further conducted by titrating trans-G with PQ ⊂ CB[8], yielding a binding constant of 9.37 ($\pm2.37$) $\times 10^4$ M$^{-1}$ (Fig. 1b), comparable to the previous reports of similar complexes[23].

**Supramolecular vesicles and PQ-loaded vesicles in water.** As CB[8] at one end serves as the hydrophilic segment and the alkyl chain moiety of trans-G located at the other side of PQ serves as the hydrophobic segment, the ternary (PQ·trans-G) ⊂ CB[8] complex is amphiphilic, and thus may self-assemble into micellar or vesicular structures. According to the poly-core theory proposed by Akiyoshi[26], in the presence of the relatively large hydrophilic domain, CB[8], the relatively short, hexyl moiety would tend to form hydrophobic microdomains and the bulky CB[8] polar head would form two hydrophilic layers surrounding these hydrophobic microdomains, yielding vesicular structures. After simple mixing of trans-G, CB[8] and PQ at a 1:1:1 stoichiometry in an aqueous solution, a vesicle structure was obtained. With the same approach and in the presence of a large excess of PQ, PQ-loaded vesicles were obtained (Fig. 2). As shown in Fig. 2a, under UV irradiation, trans-to-cis isomerization of G would break down the supramolecular vesicles and release the payload, PQ, in a controlled manner. Transmission electron microscopy (TEM) images (Fig. 2b, c) revealed the formation of hollow spheres with an average diameter of 161.4 nm, and a vesicular wall thickness of ~7 nm, consistent with the calculated thickness of the trans-G/PQ/CB[8] bilayer structure by Chem3D (~6 nm), suggesting the formation of a vesicle structure. Dynamic

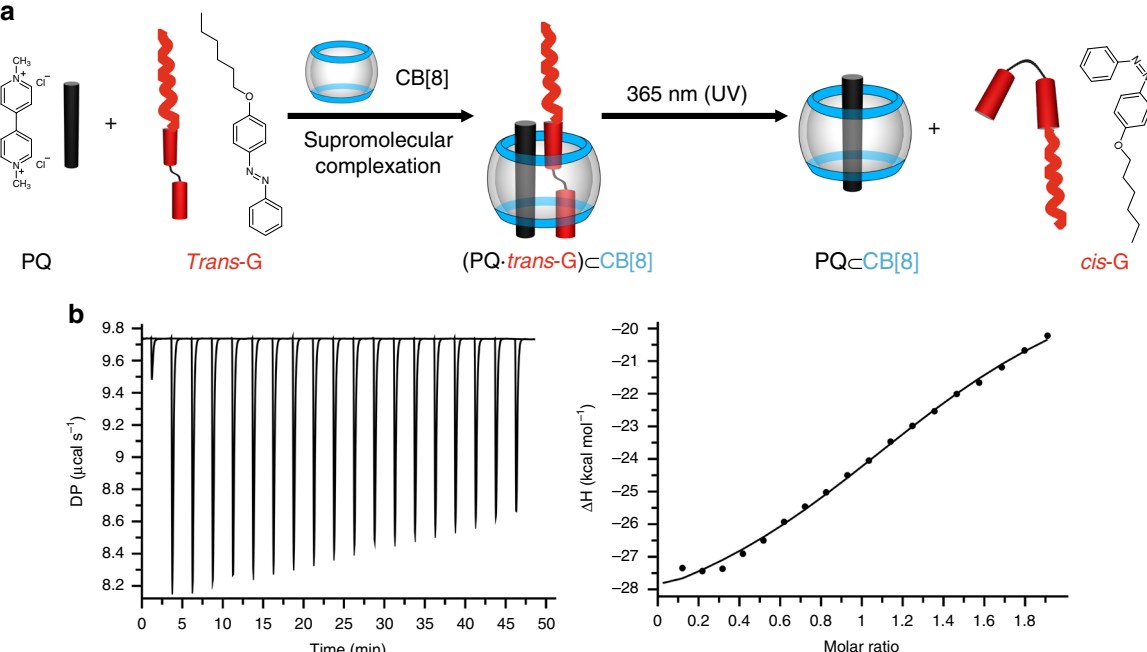

**Fig. 1** Complexation of CB[8], PQ, and *trans*-G. **a** Schematic illustration of CB[8]-mediated complexation with PQ as the first guest and *trans*-G as the second guest, as well as the photo-driven, reversible transition between complexation with the *trans*-G isomer and decomplexation with the *cis*-G isomer. **b** ITC thermograms: *trans*-G (0.04 mM) titrated with PQ ⊂ CB[8] (0.4 mM)

light scattering (DLS) analysis (Fig. 2d) of the vesicles exhibited a narrow size distribution with an average diameter of 187.8 nm. In addition, the zeta potential measurements afforded −21.7 mV (Supplementary Fig. 2), indicating that the vesicles had a good colloidal stability. After storage in PBS and DMEM containing 10% FBS, respectively, under dark conditions for 3 days, the mean diameter and polydispersity index increased very moderately, confirming their structural stability in the dark (Supplementary Table 1). The storage of these supramolecular vesicles in PBS was further extended to 210 days. Although the diameter of the vesicles increased from 208.1 to 293.5 nm, these PQ-loaded vesicles after 210-day storage still exhibited a good herbicide release profile under sunlight, as well as an excellent safety profile as evaluated on a mouse model for 31 days (details in the following sections). In addition, the stability of the vesicles in PBS with different biologically relevant pH (5.0, 5.8, 6.6, and 7.4) were evaluated for 36 h, and no changes in sizes were observed (Supplementary Fig. 3). These results suggested their excellent stability in an environment that may be encountered in vitro or in vivo.

As shown in Fig. 2a, the PQ-loaded vesicles were prepared by a facile one-pot preparation by simply mixing CB[8], excess PQ and *trans*-G. When the ratio of CB[8], PQ and *trans*-G was 1:2:1, the drug loading capacity (DLC) and drug encapsulation efficiency (DEE) were 2.2% and 16.4%, respectively.

**Photo-responsiveness of the vesicles**. Stimuli-responsive supramolecular amphiphiles that can self-assemble into nanoparticles in aqueous solution has drawn much attention in recent years due to their potential for controlled payload delivery[27,28]. Among various examples, (MV$^{2+}$·*trans*-azobenzene) ⊂ CB[8] has exhibited well-controlled photo-responsiveness in aqueous solutions[29]. Therefore, the present vesicles formed from (PQ·*trans*-G) ⊂ CB[8] may exhibit photo-responsiveness as well. As expected, the vesicles moderately collapsed from their spherical shape after exposure to UV light for several minutes, as shown in the TEM image (Fig. 2e), along with the disappearance of Tyndall effects

(Fig. 2f), and the particle size increased dramatically, as measured by DLS (Fig. 2g). We further investigated the photo-triggered herbicide release of PQ-loaded vesicles. As shown in Fig. 2h, the cumulative release ratio of PQ was only ~10% within 10 h when the PQ-loaded vesicles were placed in the dark, whereas the majority of PQ (~90%) were released within 24 min under continuous irradiation of UV light, demonstrating excellent UV responsiveness of the PQ-loaded vesicles. Under irradiation with a sunlight simulator (360–800 nm spectral output, 30 W), PQ was released gradually and a significant increase of cumulative release ratio was observed after irradiation for 5 h, reaching up to ~75% release within 10 h. As the herbicide would be used under natural sunlight conditions in the real world, it is essential to demonstrate the triggered release of PQ from the PQ-loaded vesicles under sunlight. As shown in Supplementary Fig. 4a, it took only ~4 h (much less time than that under simulated sunlight) to reach a cumulative PQ release ratio of 80% under natural sunlight. This might be attributed to the wide spectrum window (290–3200 nm) of the natural sunlight in comparison with the simulated sunlight (360–800 nm). Furthermore, diquat, a non-selective, fast-acting desiccant herbicide, was encapsulated within these photo-responsive vesicles as another model herbicide. Its herbicide release profile was similar to that of PQ-loaded vesicles under natural sunlight (Supplementary Fig. 4b), indicating that these photo-responsive vesicles provided a promising platform for sunlight sensitive release of herbicides.

**In vitro cytotoxicity and ROS generation**. With regard to possible damage on the liver and kidneys after acute PQ exposure[30], the toxicity of PQ-loaded vesicles on L-02 cell line (human liver cells) and COS-7 cell line (monkey kidney fibroblast cells) were firstly evaluated. As shown in Fig. 3a, PQ exhibited a dose-dependent toxicity to both cell lines after incubation for 24 h. In contrast, the relative viability of L-02 and COS-7 cell lines incubated with PQ-loaded vesicles in the dark were significantly better. Furthermore, after exposure to UV light, PQ-loaded

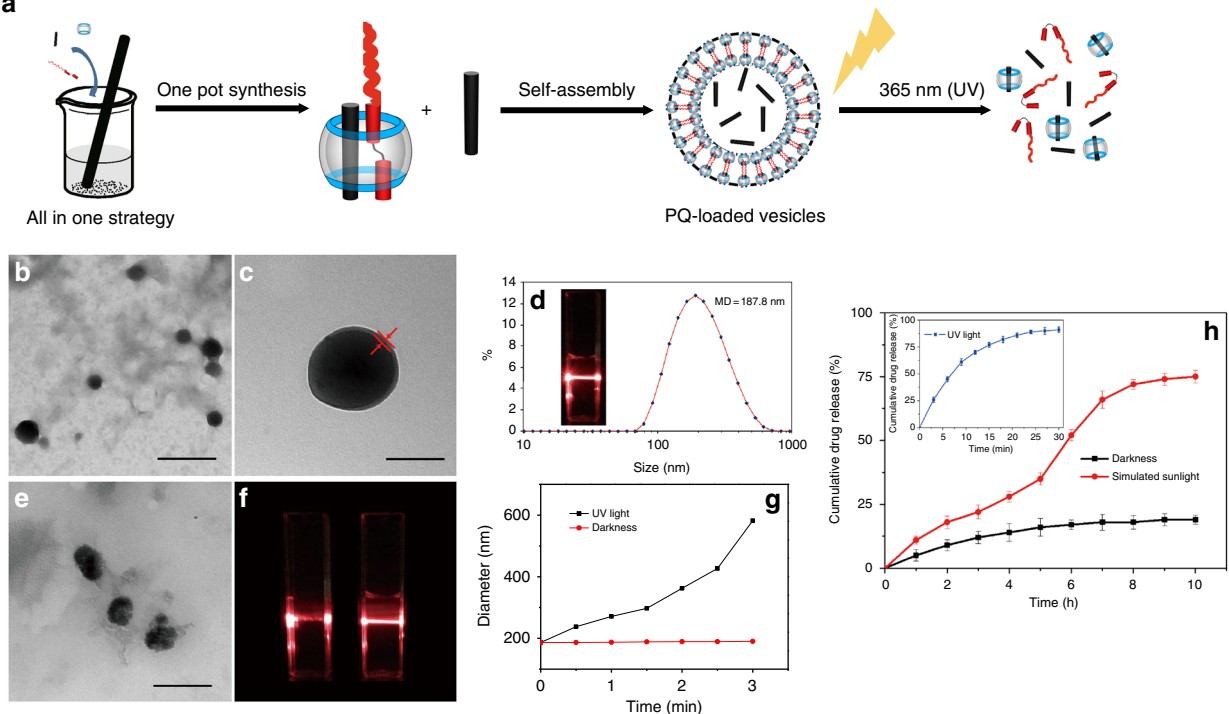

**Fig. 2** Preparation of photo-responsive vesicles. **a** Schematic illustration of the preparation and photo-responsiveness of PQ-loaded vesicles. **b** TEM image of vesicles. Scale bar: 500 nm. **c** Enlarged TEM image. Scale bar: 100 nm. **d** DLS data of vesicles. Inset: photo showing the Tyndall effect of vesicles; **e** TEM image of vesicles after exposure to UV light. Scale bar: 500 nm. **f** Tyndall effect of vesicles before (left) and after (right) exposure to UV light. **g** Diameter changes of vesicles over time in the dark or under irradiation by UV light. **h** Release profile of PQ from PQ-loaded vesicles in the dark and under irradiation by simulated sunlight. Inset: release profile of PQ from PQ-loaded vesicles under irradiation by UV light. Data was combined from three individual experiments and presented as means ± SEM

vesicles exhibited strong cytotoxic effects on both cell lines, comparable to that of free PQ. This result might be attributed to the UV-responsive breakdown of the vesicles, resulting in the release of PQ. On the other hand, the toxicity of CB[8] + *trans*-G, exhibited moderate cytotoxic effects against both of these cell lines (Supplementary Fig. 5). As PQ's toxicity was mainly due to generation of reactive oxygen species (ROS)[31], the ROS levels in both cell lines were determined after incubation with PQ and vesicle formulation of PQ, respectively, for 12 h. As shown in Fig. 3b, PQ-loaded vesicles that were incubated in the dark exhibited little effects on the generation of ROS in both cell lines, suggesting an improved safety profile of the PQ-loaded vesicles without significant exposure to light. In contrast, both free PQ and PQ-loaded vesicle formulation (under UV irradiation) increased the levels of ROS in both cell lines. Additionally, cell apoptosis rates of L-02 and COS-7 cells treated with PQ and PQ-loaded vesicles were, respectively, measured by flow cytometry. As shown in Fig. 3c–f, the apoptosis rates of both cell lines were significantly increased after treatment with PQ for 6 h. However, PQ-loaded vesicle formulation (incubated in dark) had negligible effects on the apoptosis rates of both cell lines. In contrast, under UV irradiation, the apoptosis rates induced by PQ-loaded vesicles in both cell lines were increased to the level induced by free PQ, implying that the PQ was likely released under UV irradiation.

**Cellular uptake and intracellular payload release**. With the cytotoxicity data obtained, it was of importance to understand the cellular uptake behaviors of the vesicles. Thus, L-02 and COS-7 cell lines were, respectively, incubated with Cy5-loaded vesicles. As shown in the fluorescent confocal microscopic images

(Supplementary Fig. 6), the red fluorescence of Cy5-loaded vesicles was observed in the cytoplasm with an incubation time-dependent uptake increase. Although the vesicles were internalized within the cells, the release of the payload inside the cells would dictate the cellular toxicity. Therefore, intracellular release of the payload was examined. Nile red (NR) is a lipophilic stain without fluorescence itself, whereas it emits strong red fluorescence when combined with an intracellular lipid droplet. Thus, NR-loaded vesicles were prepared to evaluate the intracellular payload release within the cells. After incubation for 2 h, almost no red fluorescence was observed in both L-02 and COS-7 cells, without irradiation by UV light (as shown in Fig. 3g), presumably because the NR was not released into the cytoplasm. This behavior indicated that the vesicles were stable inside the cellular environment, which was consistent with the previously observed stability profile of the vesicles in aqueous solutions and media (Supplementary Table 1). Upon exposure to UV light for 16 s during incubation of the cells with NR-loaded vesicles, weak red fluorescence was detected in both cell lines after incubation for 10 min, and the fluorescence intensity in the cytoplasm increased moderately over time (Fig. 3g). These results demonstrated that the vesicles had a good stability in the dark even when they were taken up by the cells, and they still exhibited UV-responsive payload release within the cells.

**In vivo safety evaluation in a zebrafish model**. The safety profile of PQ-loaded vesicles was evaluated in zebrafish, as it is a rapid and inexpensive in vivo model to evaluate the safety profile of chemical substances or formulations. Firstly, Cy7.5-loaded vesicles with red fluorescence were utilized to analyze the bio-uptake

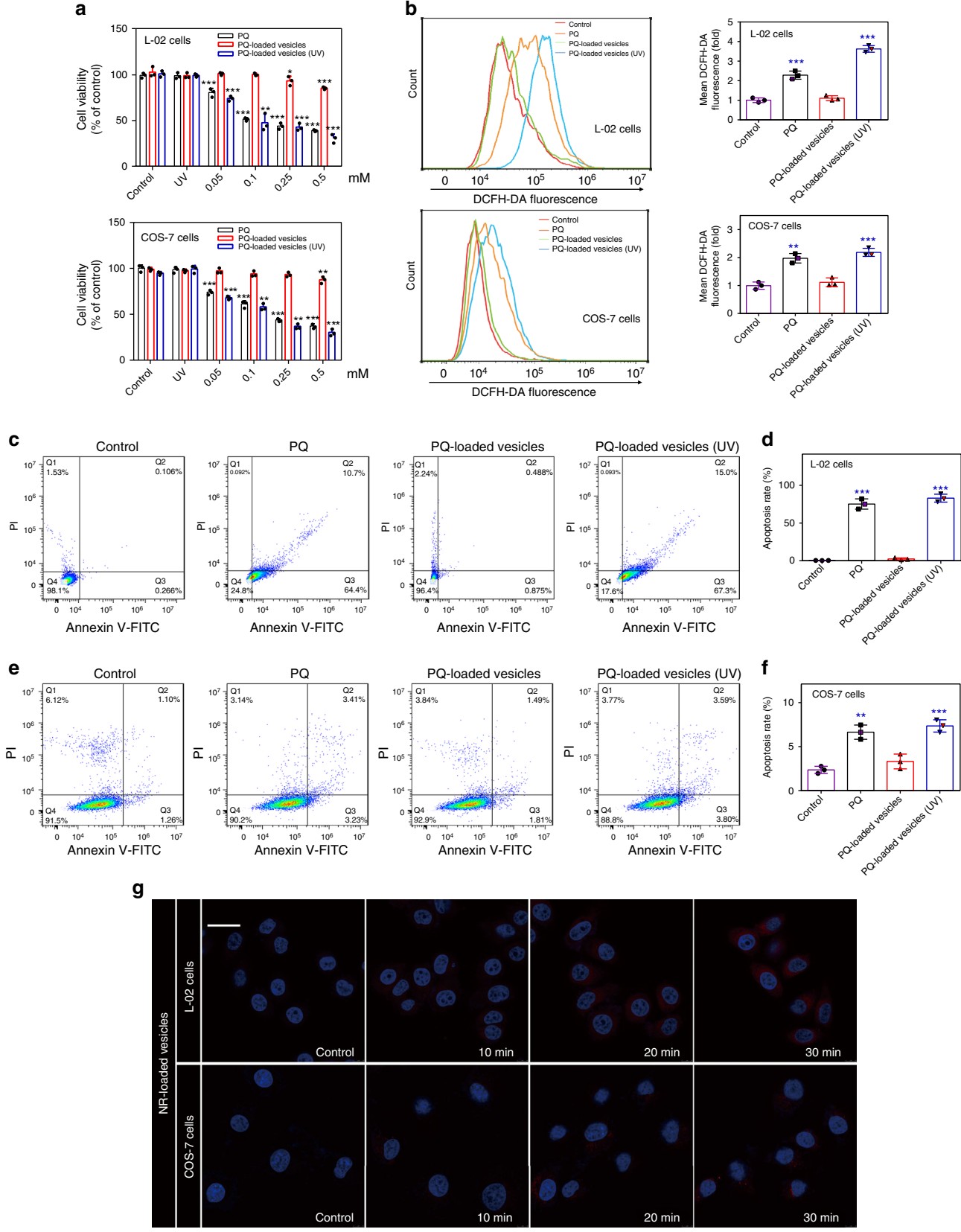

and distribution of the vesicles in zebrafish at the age of 2 dpf (days post fertilization). Figure 4a showed that the vesicles were mainly distributed in the yolk sac, liver, and kidneys, and the fluorescence intensity was increased in a dose-dependent manner (Fig. 4b). Moreover, it had been reported that the hepatotoxicity in zebrafish could be induced by PQ[32]. Upon examination under microscope, the liver of zebrafish treated with free PQ was amorphous and gray, an indicative of necrosis, whereas the zebrafish liver in the control group exhibited clear liver tissues (Fig. 4c). Meanwhile, the darkness index of the liver area was relatively small in the zebrafish incubated with PQ-loaded vesicles in the dark. Conversely, a darker index in the liver was observed in the zebrafish treated with PQ-loaded vesicles with brief UV irradiation. These results were semi-quantitatively analyzed (Fig. 4d), and confirmed that the hepatotoxicity of the PQ-loaded vesicles-treated group was significantly reduced in comparison to the PQ-treated group.

Additionally, older zebrafish embryos, at the age of 30 dpf, known to orally take up nutrients and substances in the medium[33], were employed to further evaluate the toxicity of PQ-loaded vesicles. Zebrafish were treated with PQ and PQ-loaded vesicles (2 mM, with or without UV irradiation) for different time lengths (up to 60 h), respectively. As shown in Supplementary Table 2, the PQ concentration in zebrafish treated with PQ-loaded vesicles was slightly higher than that in free PQ-treated zebrafish, which could be attributed to the excellent penetration ability of nanosized materials that may facilitate transdermal uptake of PQ in addition to oral uptake. The PQ uptakes in the zebrafish treated with PQ-loaded vesicles without and without UV irradiation are comparable. As shown in Fig. 4e, nearly 90% of these 30-dpf zebrafish treated with PQ-loaded vesicles (in the dark) survived after 60 h incubation. In contrast, brief UV irradiation (~2 min) of the zebrafish treated with PQ-loaded vesicles led to 100% mortality within 50 h of incubation, similar to that of zebrafish treated by free PQ. Although bio-uptakes of PQ-loaded vesicles with or without UV irradiation were nearly identical, their significantly different mortality rates further confirmed a decent safety profile of PQ-loaded vesicles in zebrafish that were not exposed to UV light.

**In vivo safety evaluation in a mouse model.** A mouse model was employed to further examine and verify the safety profile of the vesicles. In this study, initially, 6-week-old female C57BL/6 mice were intraperitoneally administered a single-dose of PQ at 20, 30, 40, and 50 mg kg$^{-1}$ levels on the first day, and their behaviors as well as their survival rates were subsequently monitored for dose-escalation evaluation. As shown in Supplementary Fig. 7a, the mice generally exhibited dose-dependent survival rates. Therefore, a low dose of 20 mg kg$^{-1}$ and a relatively high dose of 40 mg kg$^{-1}$ PQ doses were selected for, respectively, evaluating body-weight changes and overall survival rate for 31 days after PQ or PQ-loaded vesicles administration. As shown in Fig. 5a, in contrast to PQ-treated mice that experienced remarkable weight loss during the first two days, the mice administered with the

vesicles loaded with PQ (20 mg kg$^{-1}$) experienced normal body-weight growth that was comparable with that of the control group. In a separate set of experiments regarding the overall survival rate (Fig. 5b), ~66% of the mice administered with free PQ (40 mg kg$^{-1}$) died on the fourth day post PQ-administration. In contrast, the survival rate on the 31th day post PQ-administration was 100% for the mice administered with PQ-loaded vesicles. These results demonstrated that the PQ-loaded vesicles are much safer to mammals than free PQ.

Further toxicity examinations on major organs in mice administered with 30 mg kg$^{-1}$ dose of PQ and PQ-loaded vesicles were conducted, respectively. The liver is the main site for xenobiotic metabolism and is highly vulnerable to damage induced by PQ due to the generation of ROS[34,35]. The liver function biomarkers test on both day 3 and day 31 post administration showed that the levels of alanine transaminase (ALT) and aspartate aminotransferase (AST) in the plasma of the mice treated with free PQ were increased remarkably and remained a high level (Fig. 5c), suggesting a possible long-term damage of the liver, consistent with previous reports[35]. In contrast, the mice treated with PQ-loaded vesicles maintained their liver function, as shown by the levels of ALT and AST, similar to those of the control group (Fig. 5c). Additionally, to evaluate potential damage on the kidneys, the quantities of uric acid (UA) and blood urea nitrogen (BUN) in the serum, as common clinical biomarkers of the renal function, were analyzed. As shown in Fig. 5d, UA in the mice treated with PQ increased on both day 3 and day 31 post administration, suggesting a long-term kidney damage by PQ. In contrast, the UA level in the mice treated with PQ-loaded vesicles was nearly identical to that in the control group. Meanwhile, either PQ or PQ-loaded vesicles did not affect the levels of BUN in the serum of the mice, suggesting that the renal function was likely preserved in part. Additionally, as chronic PQ exposure may lead to Parkinson's disease (PD)[36], PD-related biomarkers, neurotransmitter (dopamine, DA) and metabolite (dihyrophenylacetic acid, DOPAC)[37], in the striatum of the mice were analyzed by liquid chromatography-mass spectrometry (LC-MS). The results (Fig. 5e) demonstrated that the levels of DA and DOPAC were not affected in all the groups of mice. This observation is consistent with previous observation that PQ could not cross the blood brain barrier to a significant extent in single-dose treatment[38]. In addition, PQ poisoning has reportedly caused severe damage to the lungs, which was indicated by increased levels of inflammation cytokines[39]. Figure 5f showed that the levels of interleukin-1β (IL-1β), tumor necrosis factor-α (TNF-α), IL-6, and IL-10 in the lungs of the mice treated with free PQ were increased significantly on both day 3 and day 31 post administration, suggesting a long-term damage of the lungs. Conversely, the levels of these inflammation cytokines in the mice treated with PQ-loaded vesicles were comparable with those of the control group. Taken together, these results further demonstrated the low toxicity of PQ-loaded vesicles in vivo.

The toxicity of PQ and PQ-loaded vesicles on these major organs of mice was further evaluated by examining tissue

---

**Fig. 3** Safety evaluation of PQ-loaded vesicles in cell model. **a** Cytotoxicity evaluation of PQ and PQ-loaded vesicles incubated in L-02 cells and COS-7 cells before and after exposure to UV light. Error bar corresponds to the SD ($n = 3$). *$P \leq 0.05$, **$P \leq 0.01$, and ***$P \leq 0.001$ determined by two-way ANOVA followed by Dunnett's test. **b** Intracellular ROS in L-02 cells and COS-7 cells. Apoptosis rates (measured by flow cytometry) and quantitative analysis on the apoptosis rates of **c**, **d** L-02 cells, and **e**, **f** COS-7 cells, induced by PQ and PQ-loaded vesicles under different conditions of light exposure. The error bar corresponds to the SD ($n = 3$). *$P \leq 0.05$, **$P \leq 0.01$, and ***$P \leq 0.001$ determined by one-way ANOVA followed by Dunnett's test. **g** Intracellular payload release of NR-loaded vesicles in L-02 cells and COS-7 cells before (control group) and after exposure to UV light for 16 s after co-incubation for 10, 20, and 30 min. Scale bar: 100 μm

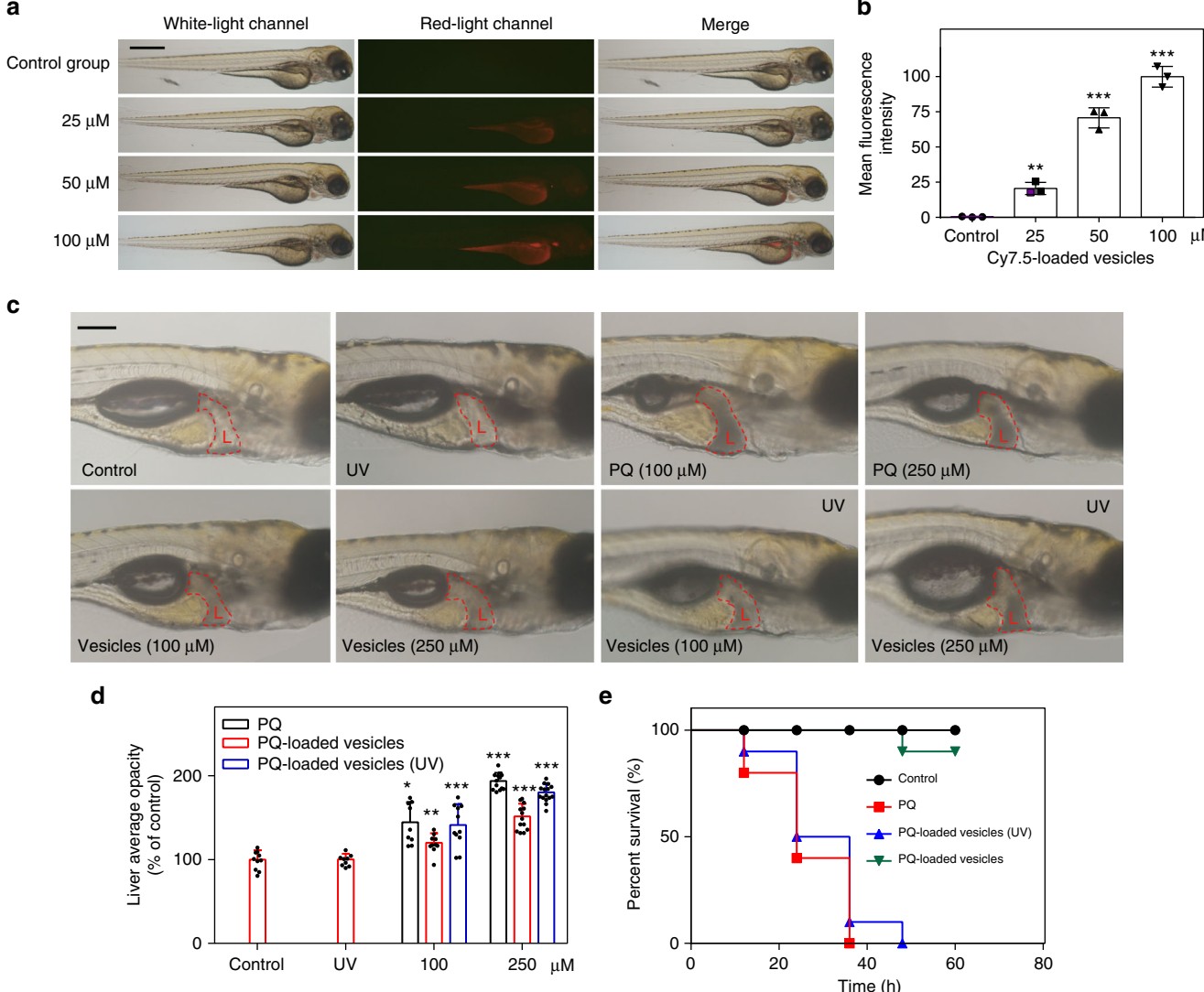

**Fig. 4** Safety evaluation of PQ-loaded vesicles in zebrafish model. **a** Two-dpf zebrafish incubated with media containing Cy7.5-loaded vesicles of various concentrations for 2 days. Scale bar: 0.6 mm. **b** Semi-quantitative analysis of bio-uptake of Cy7.5-loaded vesicles. The error bars correspond to the SD ($n = 3$). *$P \leq 0.05$, **$P \leq 0.01$, and ***$P \leq 0.001$ determined by one-way ANOVA; **c** 2-dpf zebrafish incubated with media containing PQ-loaded vesicles with different concentration for 2 days. The red-dotted region refers to the liver tissue. Scale bar: 0.2 mm. **d** Semi-quantitative analysis of liver average opacity. The error bars correspond to the SD ($n = 9$–15). *$P \leq 0.05$, **$P \leq 0.01$, and ***$P \leq 0.001$ determined by Student's $t$ test. **e** Thirty-dpf zebrafish incubated with PQ and PQ-loaded vesicles (2 mM), respectively, for 60 h, with or without UV irradiation ($n = 10$ in each group)

histology, organ index, and PQ distribution. As shown in Fig. 5g (from samples collected on day 3) and Supplementary Fig. 7b (from samples collected on day 31), hematoxylin and eosin (H&E)-stained liver tissue from the free PQ-treated mice showed obvious signs of hemorrhage at the central vein, atrophy of the hepatocytes around the central vein, and parenchymal inflammation. Similarly, the lung tissues from this group of mice also exhibited severe inflammation, hemorrhage, and widespread thickening of alveolar septum. The kidney tissues of this group exhibited obvious signs of swelling, vacuolar degeneration, and interstitial hyperemia. Conversely, the pathological changes of the organs from the mice treated with PQ-loaded vesicles were significantly attenuated. Meanwhile, no pathological changes were observed in the striatum and nigra from both PQ group and PQ-loaded vesicles groups, indicating that the brain was free of damage from a single-dose administration, consistent with previous report[38] and PD biomarkers analysis (Fig. 5f). In order

to evaluate possible fibrosis conditions of the lungs, Masson's trichrome staining of lung tissue was conducted (Fig. 5h and Supplementary Fig. 7c). The lung tissue of the PQ-treated group of mice exhibited excess fibroblast proliferation and more collagen than that from the control group. In contrast, the histology and collagen content shown in the lung tissue of the mice from PQ-loaded vesicles group were similar to those of the control group.

In addition, the organ weight/body-weight indexes in major organs of the mice were evaluated. As shown in Supplementary Fig. 7d, the mean organ indexes of both the liver and kidneys of the mice treated with free PQ decreased moderately and the mean weight index of the lungs of these mice increased significantly, when compared with those of the control group. These changes were observed on both day 3 and day 31, suggesting likely irreversible damages caused by free PQ on these organs. In contrast, these organ indexes of the mice

treated with PQ-loaded vesicles were nearly identical with those of the control group. With regard to the brain, the brain weight index in all the groups did not experience significant changes.

Finally, the PQ concentrations in these organs were, respectively, analyzed by LC-MS in order to understand its biodistribution and clearance kinetics. As shown in Supplementary Table 3, PQ was mainly distributed in the liver, kidneys, and lungs of the

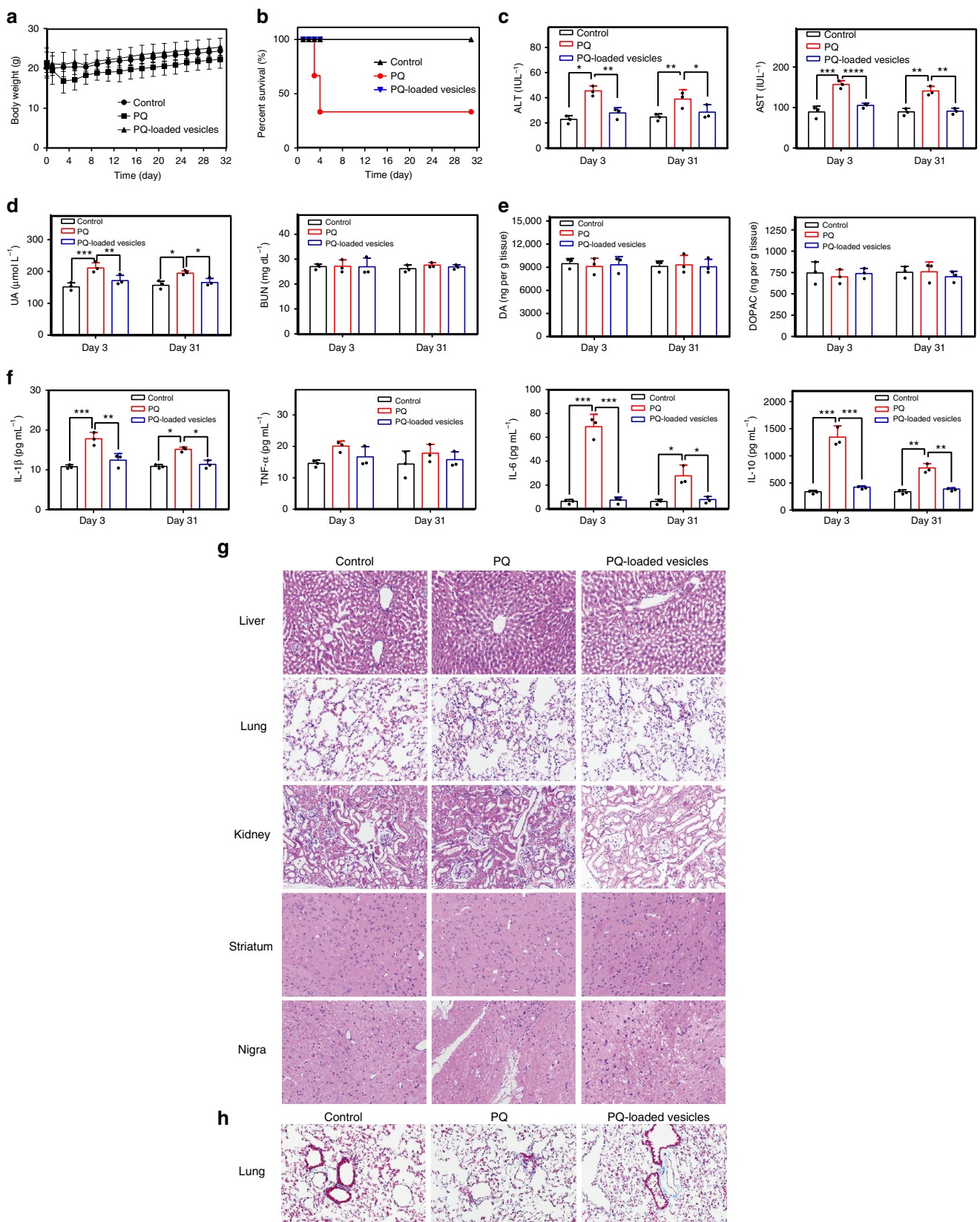

mice administered with either PQ or PQ-loaded vesicles, at 30 min post administration, without any PQ detectable in the brain. After administration for 7 days, PQ was not detectable in any of these organs in both groups of the mice, implying that both PQ and PQ-loaded vesicles were likely cleared out by this time point. Additionally, the results of whole-body fluorescence imaging of the mice administered with Cy7.5-loaded vesicles further confirmed that the payload was mostly cleared out (likely via kidneys) of the mice at the 6th day post administration (Supplementary Fig. 8). Although PQ-loaded vesicles were absorbed and distributed in different organs including the liver, kidneys, and lungs, at nearly equal level to those of the free PQ-treated group, they did not induce any noticeable damages to the major organs of the mice.

In reality, acute and chronic PQ poisonings often take place via oral ingestion and respiratory/dermal exposures, respectively, whereas our investigations were conducted in mice via intraperitoneal administration. In fact, a variety of administration routes in mice or rats have been previously validated for toxicity evaluations of PQ and PQ formulations, including oral, subcutaneous, dermal, inhalation, and intraperitoneal routes[40–42]. In our head-to-head comparative studies, PQ-loaded vesicles exhibited a significantly improved safety profile in the mouse model via intraperitoneal administration, as evaluated at molecular, tissue, organ and survival and physical growth levels, suggesting that this PQ formulation may provide next-generation, user-friendly herbicides.

**Comparative safety evaluation of PQ:LAS in vivo.** As previously reported by Dinis-Oliveira and co-workers, an optimal formulation of PQ:LAS (molar ratio of 1:2) exhibited an excellent safety profile in Wistar male rats orally administered with this formulation at PQ dose of 125 mg kg$^{-1}$[13]. In order to compare the safety profile of this literature formulation with that of our PQ-loaded vesicles in our studies, PQ:LAS (1:2) was evaluated in parallel with PQ-loaded vesicles in both zebrafish and mouse models. As shown in Supplementary Fig. 9a, all of the 30-dpf zebrafish incubated in media containing PQ:LAS (2 mM:4 mM) died within 18 h of incubation. Similarly, in three groups of mice that were intraperitoneally injected with PQ:LAS (1:2) doses (20:50, 30:76, and 40:101 mg kg$^{-1}$ PQ:LAS for body-weight changes, organ histology, and survival rates, respectively) for subsequent 31-day follow-up, as shown in Supplementary Fig. 9b–e, PQ:LAS (1:2) formulation exhibited similar or even worse toxicity profile than that of free PQ (Fig. 5a, b, g, h) in this mouse model. In fact, Dinis-Oliveira and co-workers previously reported in a separate study that a high dose of LAS may induce even more serious toxicity than PQ alone[43]. Therefore, the higher toxicity of PQ:LAS in our studies were likely caused by a relatively high-dose LAS for these zebrafish/mice models (4 mM in the zebrafish study and up to 101 mg kg$^{-1}$ in the mice study).

**Herbicidal activity.** With sufficiently demonstrated safety profile of the PQ-loaded vesicles in cellular, zebrafish, and mouse models, we moved to the investigation of its herbicidal activity. *Estuca arundinacea* is a species of invasive grass commonly known as tall fescue and the photosynthesis occurs throughout its leaves and bunches. Thus, PQ would exhibit a good weed control efficacy through interrupting the photosynthesis pathway and *E. arundinacea* was chosen as a plant model for the evaluation of herbicidal activity. Firstly, the herbicidal efficacy of PQ-loaded vesicles was studied in the simulated sunlight. After cultivation with PQ-loaded vesicles (2 mg/mL PQ dose sprayed on the grass) under irradiation with simulated sunlight in the absence of UV-range light for 120 h, the grass exhibited very little differences from the control group (Fig. 6). Upon exposure to simulated sunlight (that contained UV light), the herbicidal activity of PQ-loaded vesicles was improved significantly and demonstrated similar herbicidal effects to that of free PQ at the same dose with slightly delayed effects, as the first sign of necrosis was observed at ~48 h post administration, as shown in Fig. 6. Both groups of grass administered with free PQ and PQ-loaded vesicles showed signs of complete necrosis and dryness at ~120 h post administration. The slightly delayed herbicidal effects of PQ-loaded vesicles might be attributed to a relatively slow disassembly process of the vesicles for PQ release that took at least 7–8 h under a simulated sunlight irradiation, which was previously verified in the photo-triggered PQ release profile studies of the PQ-loaded vesicles (Fig. 2h).

Subsequently, in order to examine if the system would work as an effective herbicide in a natural setting, the herbicidal experiment was conducted under natural sunlight. The herbicidal results and associated weather conditions were shown in Supplementary Fig. 10 and Supplementary Note, respectively. The herbicidal activity of PQ-loaded vesicles was similar to that of free PQ, under sunlight in a natural setting. Very interestingly, PQ-loaded vesicles exhibited almost no delay on the herbicidal efficacy in comparison with free PQ, as they were more sensitive to the natural sunlight and released PQ quickly within ~3–4 h (Supplementary Fig. 4a), demonstrating its significant potential for practical application in green agriculture.

The cost of raw materials required for preparing PQ-loaded vesicles was listed in Supplementary Table 4 for estimation of the overall cost of this novel herbicide formulation prepared in a laboratory setting. With the current DLC of PQ-loaded vesicles being ~2.2%, the cost of PQ-loaded vesicle was approximately seven times that of free PQ. If the raw material cost can be reduced (for instance, more efficient synthetic methods of CB[8]

**Fig. 5** Safety evaluation of PQ-loaded vesicles in mouse model. **a** Body-weight changes of 6-week-old female C57BL/6 mice that were intraperitoneally administered saline as a control group and a single-dose of PQ or PQ-loaded vesicles at 20 mg kg$^{-1}$ dosage on the first day (*n* = 6 in each group). **b** Survival curve of 6-week-old female C57BL/6 mice that were intraperitoneally administered a single dose of saline, or PQ and PQ-loaded vesicles at 40 mg kg$^{-1}$ dosage on the first day (*n* = 6 in each group). **c** Quantitative analysis of liver function biomarkers (ALT and AST), and **d** renal function biomarkers (UA and BUN), analyzed in the blood of the mice intraperitoneally administered with PQ and PQ-loaded vesicles at a dose of 30 mg kg$^{-1}$. The blood samples were collected on day 3 and day 31 after dose administration, respectively. **e** Quantitative analysis of DA and DOPAC in the striatum of the mice to evaluate potential PD development, and **f** quantitative analysis of inflammatory factors (IL-1β, TNF-α, IL-6, and IL-10) in the lungs of the mice. The organ tissues were collected on day 3 and day 31 after dose administration, respectively. The error bars correspond to the SD (*n* = 3). *$P \leq 0.05$, **$P \leq 0.01$ and ***$P \leq 0.001$ determined by two-way ANOVA followed by Tukey's test. **g** H&E staining of the liver, lungs, kidneys, striatum, and nigra and **h** Masson's trichrome staining of the lungs, in the mice after intraperitoneal administration of PQ and PQ-loaded vesicles at 30 mg kg$^{-1}$ dosage. The organ samples were collected on day 3 post administration

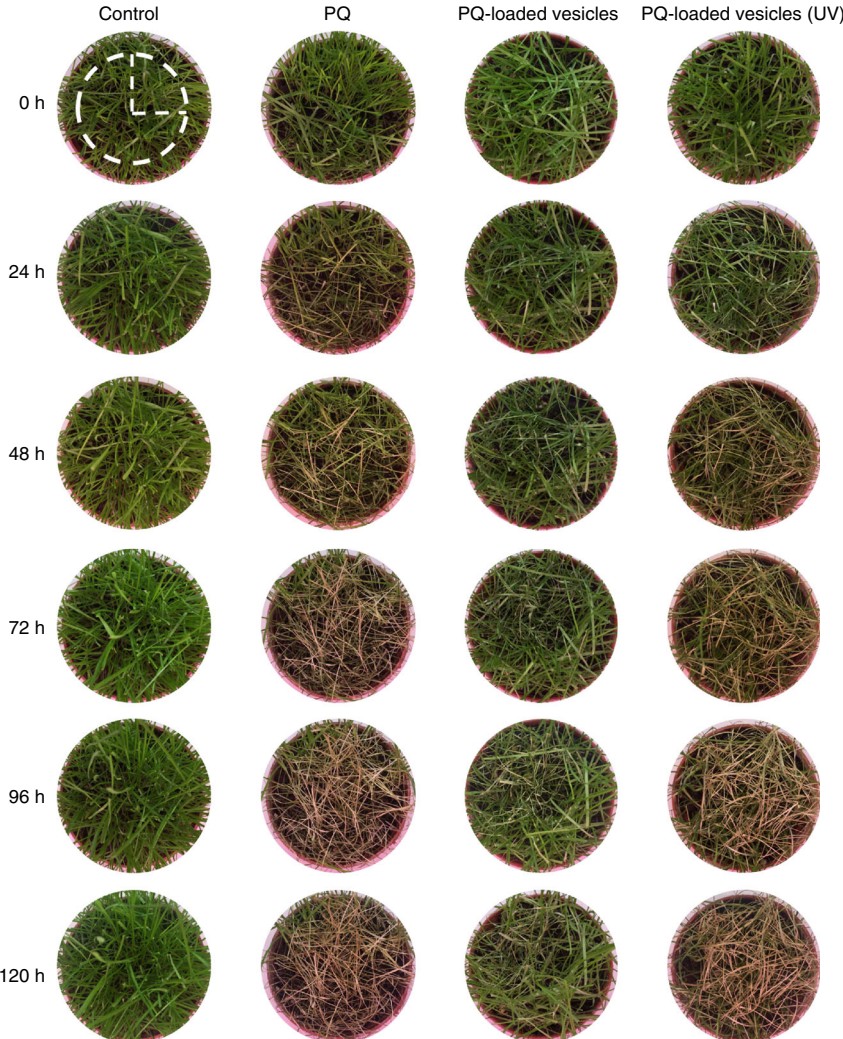

**Fig. 6** Weed control efficacy of free PQ and PQ-loaded photo-responsive vesicles. Foliar application of control (water), PQ and PQ-loaded vesicles under irradiation of simulated sunlight, and PQ-loaded vesicles under irradiation of simulated sunlight without UV light, at a single dose of 2 mg mL$^{-1}$. Diameter of the white-dashed circle: 6 cm

are developed) and the DLC becomes improved, the cost of this user-friendly herbicide would be decreased dramatically, yielding a more economic, user-friendly herbicide.

## Discussion

A photo-responsive vesicle was prepared via our one-pot synthesis by the self-assembly of amphiphilic ternary supramolecular complexes between CB[8], PQ, and hydrophobic *trans*-G. PQ not only served as a guest inside the cavity of CB[8], but also was encapsulated in the core of the vesicles as a herbicide payload. Another guest molecule of CB[8], *trans*-G, an azobenzene derivative, may respond to UV light and undergo *trans* to *cis* isomerization, and the *cis*-isomer would dissociate from the ternary complex and break down the vesicles, so that the payload would be released.

Due to the sunlight-responsive payload release, this PQ-loaded vesicle exhibited an excellent safety profile on cellular, zebrafish, and mouse models, in contrast to the inherently toxic free PQ. Additionally, the herbicidal activity of PQ-loaded vesicles against *E. arundinacea* (a model invasive weed) was nearly identical to that of free PQ under natural sunlight. Therefore, these sunlight-responsive vesicles provide a platform to turn highly toxic

herbicides or pesticides into user-friendly formulations that have been desired for decades in green agriculture.

## Methods

**Materials**. All chemical reagents were purchased from Sigma Aldrich (USA). CB [8] was synthesized via a literature procedure[44]. Cell culture medium was supplied by Gibco (USA).

The normal human liver cell line L-02 and the monkey kidney cell line COS-7 were purchased from ATCC (Shanghai, China), where they were authenticated by DNA fingerprinting, mycoplasma detection, isozyme detection, and cell vitality test. Wild-type zebrafish were used for the in vivo bio-uptake study and hepatotoxicity of PQ-loaded vesicles. Female C57BL/6 mice (6 weeks) were purchased from Faculty of Health Sciences, University of Macau. All of the animal procedures were approved by the Animal Ethics Committee, University of Macau.

**Synthesis of G**. A solution of 4-phenylazophenol (1.982 g) and potassium carbonate (2.073 g) in DMF (25 mL) was stirred for 0.5 h at room temperature. Bromohexane (2.135 mL) was subsequently added to the solution and the mixture was stirred for 24 h at room temperature. After removal of the solvent by reduced pressure distillation, the crude product was purified via silico gel column chromatography to give pure yellow powder (yield 92.16%). $^{1}$H NMR (600 MHz, DMSO-$d_6$): δ 7.86 (d, 2 H), 7.82 (d, 2 H), 7.55 (t, 2 H), 7.50 (t, 1 H), 7.1 (d, 2 H), 4.06 (t, 2 H), 1.73 (m, 2 H), 1.42 (m, 2 H), 1.30 (m, 4 H), 0.87 (t, 3 H). $^{13}$C NMR (600 MHz, DMSO-$d_6$): δ 162.18, 153.23, 146.48, 131.29, 129.84, 125.84, 123.47, 115.75, 68.76, 32.22, 29.13, 26.58, 23.75, 14.43 (Supplementary Fig. 11).

**Preparation of vesicles and PQ-loaded vesicles**. CB[8] (2.5 μmol) and PQ (2.5 μmol) were dissolved in 50 mL of deionized water. Subsequently, 2.5 μmol of G in 500 μL of ethanol was subsequently added into the solution and the mixture was stirred for 4 h. The resultant solution was filtered through a filter (0.45 micron pore size) to remove any solids and a vesicle solution was obtained. After lyophilization, the obtained powders were stored for future use. The morphology of the vesicles was studied by a TEM (H-7650, Hitachi Ltd.). The particle size and distribution were measured via DLS with a Zetasizer (Nano-ZS, Malvern) system.

For the PQ-loaded vesicles, PQ was not only encapsulated in the core of the vesicles but also served as a guest in cavity of CB[8]. The process for preparing PQ-loaded vesicles is similar to that of vesicles. CB[8] (2.5 μmol) and PQ (5 μmol) were dissolved in 50 mL of deionized water, and G (2.5 μmol) in 500 μL of ethanol was added. After stirring for 4 h, the resultant solution was dialyzed in a dialysis bag (molecular weight cut-off = 12,000) to remove the unencapsulated PQ. Subsequently, the solution was passed through a filter and lyophilized for future use. The total amount of PQ in the PQ-loaded vesicles was measured by high-performance liquid chromatography (HPLC) with a reversed-phase column (Agilent TC-C18, 4.6 × 250 mm, 5 μm) at a detection wavelength of 257 nm. The mobile phase consisted of 10 mM of sodium heptanesulfonate and methanol (1:1, V%). The DEE and DLC were calculated by using Eqs. (1) and (2), respectively.

$$DEE\,(\%) = \frac{\text{Mass of drug in vesicles} - \text{Mass of drug bound with CB[8]}}{\text{Mass of drug in feed} - \text{Mass of drug bound with CB[8]}} \times 100\%, \tag{1}$$

$$DLC\,(\%) = \frac{\text{Mass of drug in vesicles} - \text{Mass of drug bound with CB[8]}}{\text{Mass of vesicles}} \times 100\%. \tag{2}$$

**Measurement of photo-sensitivity and light triggered release**. The photo-sensitivity of the vesicles was measured by DLS under different conditions of light exposure. Briefly, a dry powder of vesicles was dissolved in deionized water and divided into two aliquots. One aliquot was the control group and kept in the dark. The other aliquot was irradiated with UV light (365 nm) for 10 min. The particle sizes were determined at different time points.

The release profiles of PQ from PQ-loaded vesicles were studied in the dark, under the irradiation of UV light (365 nm, 15 W radiant output), simulated sunshine (360–800 nm spectral output, 30 W radiant output), and natural sunlight (Macau, 3 March 2018, Sunny), respectively. PQ-loaded vesicles (10 mg) in 4 mL of deionized water were placed in a dialysis bag (molecular weight cut-off = 12000) that was then transferred into 40 mL of deionized water in a shaker under different light exposure conditions. At pre-determined time intervals, 4 mL of each of incubated media was collected for chromatographic analysis and 4 mL of fresh medium was supplemented. The cumulative release of PQ was calculated using Eq. (3):

$$\text{Cumulative drug release}\,(\%) = \frac{Mt}{M0 - M0CB} \times 100\%, \tag{3}$$

where Mt denotes the amount of drug released at time t, M0 is the initial amount of drug in the vesicles, and M0CB is the amount of drug bound with CB[8]. The experiments were conducted in triplicate.

**In vitro safety evaluation in cells**. The in vitro cytotoxicity of PQ and PQ-loaded vesicles under different conditions of light exposure were determined towards L-02 cells and COS-7 cells by MTT assays. The cell viability was evaluated under a fluorescent microscope.

The L-02 cells and COS-7 cells were incubated in the media containing PQ and PQ-loaded vesicles for 12 h, respectively. Subsequently, the media were replaced with fresh media containing 2′,7′-dichlorofluorescin diacetate (DCFH-DA). After incubation for 30 min, the cells were washed with PBS for three times and the fluorescence intensity of dye cells was assessed by flow cytometry at excitation wavelength of 488 nm.

To prepare the apoptosis assay, L-02 cells and COS-7 cells were seeded, respectively, in a 12-well plate allowing incubation for 24 h. Subsequently, the media were replaced by fresh media containing PQ and PQ-loaded vesicles, respectively, allowing additional incubation for 6 h. After washing twice with PBS buffer, the cells were suspended in 100 μL of binding buffer, and were subsequently mixed with 10 μL of annexin V-fluorescein isothiocyanate (V-FITC) and 10 μL of propidium iodide for 15 min. Another 400 μL of binding buffer was added and the cells were analyzed by a flow cytometer to determine the apoptosis rates.

**Cellular uptake and intracellular drug release**. The cellular uptake and intracellular drug release tests were carried out in L-02 cells and COS-7 cells. Because PQ-loaded vesicles themselves exhibit no fluorescence, cyanine5 (Cy5), a red fluorescence dye, was loaded into the vesicles (Cy5-loaded vesicles) according to the protocol of preparing PQ-loaded vesicles. The L-02 cells and COS-7 cells were seeded in confocal dish at a density of $4 \times 10^4$ mL$^{-1}$. After incubation at 37 °C for 24 h, the media were replaced with fresh media containing Cy5-loaded vesicles, and

incubated for 1, 2, 3, and 4 h. The cells were subsequently washed for three times with PBS and fixed by paraformaldehyde for 15 min. Subsequently, the cells were washed with PBS for three more times and counterstained with Hoechst for 15 min. After washing with PBS for three times again, the cells were observed via confocal laser scanning microscopy (TCS SP8, Leica) with an excitation wavelength at 649 nm.

For the intracellular drug release behavior, NR, another fluorescence probe, was loaded into the vesicles (NR-loaded vesicles). L-02 cells and COS-7 cells were incubated in the media containing NR-loaded vesicles for 2 h and the media were replaced with fresh media. The cells were subsequently irradiated with UV light for 16 s and imaged via microscopy with an excitation wavelength at 480 nm after continuous incubation for 10, 20, and 30 min, respectively.

**In vivo safety evaluation in zebrafish**. Wild-type zebrafish provided a rapid and economical model for the safety evaluation of PQ-loaded vesicles. All embryos were raised in E3 media (5 mM NaCl, 0.17 mM KCl, 0.33 mM CaCl$_2$, 0.33 mM MgSO$_4$ at pH 7.2–7.3)[45]. For the bio-uptake of PQ-loaded vesicles, cyanine7.5 NHS ester (Cy7.5) was encapsulated into vesicles (Cy7.5-loaded vesicles). Two days post fertilization (dpf), zebrafish dechorionated manually were randomly and investigator-blindly separated into 24-well microplates (10 fish per well). Subsequently, 2-dpf zebrafish were treated with the pure medium and Cy7.5-loaded vesicles with different concentrations (25, 50, and 100 mM). After incubation for 2 days, the uptake of Cy7.5-loaded vesicles was measured using an Olympus DSU (Disk Scanning Unit) Confocal Imaging System. Subsequently, the dose of PQ and PQ-loaded vesicles was reduced to 100 and 250 mM, and the hepatotoxicity induced by PQ was measured by a confocal microscopy imaging system. The red fluorescent intensity in zebrafish was determined by the Image J version 1.49 software package for semi-quantitative analysis of Cy7.5-loaded vesicles. For the survival experiment, 30-dpf zebrafish were incubated in the media containing PQ and PQ-loaded vesicles (2 mM PQ equivalent dose), respectively, under different light exposure conditions. Finally, the concentration of PQ in 30-dpf zebrafish after incubation for 1, 2, 4, 6, and 8 h, with and without UV exposure, were, respectively, analyzed by LC-MS. The zebrafish incubated with pure media were employed as a control group.

**In vivo safety evaluation in mice**. As an initial dose-escalation study, female C57BL/6 mice (6-week old) were intraperitoneally administered with PQ at doses of 20, 30, 40, and 50 mg kg$^{-1}$, respectively, on the first day and their survival rates were examined over a period of 10 days ($n = 6$ in each group). In each set of studies, the mice were randomly and investigator-blindly divided into different groups with six mice in each group, including a control group (saline), a PQ-treated group, a PQ-loaded vesicle-treated group, and PQ:LAS-treated group. A dose of 20 mg kg$^{-1}$ was chosen to evaluate the changes of body weight of the mice after a single-dose administration of saline, PQ, PQ-loaded vesicles, and PQ:LAS (1:2), respectively. Meanwhile, a dose of 30 mg kg$^{-1}$ was selected for evaluations of the toxicity/function of major organs of the mice. In this set of experiments, blood was collected from the mice treated with saline, PQ, and PQ-loaded vesicles, respectively, for testing of the hepatic damage biomarkers (ALT and AST) and renal function biomarkers (UA and BUN), after dose administration for 3 and 31 days, respectively. The livers, lungs, kidneys, and brains were collected on day 3 and 31 after dose administration, respectively, and stored at 4 °C. The inflammatory factors in the lungs (TNF-α, IL-1β, IL-6, and IL-10) were determined by ELISA (eBiocience). With regard to the brain, the PD-related neurotransmitter (DA) and metabolite (DOPAC) in the striatum were measured by LC-MS. Hematoxylin and eosin staining of all these organ tissues and Masson's trichrome staining of the lung tissues were conducted according to the standard method[46].

In addition, after dose administration with saline, PQ, and PQ-loaded vesicles for 30 min and 7 days, respectively, the liver, lungs, kidneys, and brain of mice were collected and the PQ distribution in these organs were measured by LC-MS ($n = 6$ in each group). The PQ dose was elevated to a higher level, 40 mg kg$^{-1}$, for overall survival evaluation for 31 days as another set of experiments.

**Herbicidal activity**. Herbicidal activity of PQ-loaded vesicles was assessed in a grass lawn of *E. arundinacea*. The seeds of the lawn were cultivated in flowerpots at $24 \pm 0.5$ °C with a humidity of $70 \pm 6\%$, and illuminated with simulated sunlight (16:8 h light/dark cycle). After cultivation for 2 weeks, the flowerpots were randomly and investigator-blindly divided into eight groups (three flowerpots for each group). Firstly, the herbicidal evaluation under simulated sunlight was conducted. The four groups were treated as follows: (1) control group, grass received only water; (2) PQ group, grass exposed to PQ; (3) PQ-loaded vesicles group, grass exposed to PQ-loaded vesicles under irradiation by simulated sunlight without UV (410–800 nm spectral output, 30 W radiant output); (4) PQ-loaded vesicles group (UV), grass exposed to PQ-loaded vesicles under irradiation by simulated sunlight (360–800 nm spectral output, 30 W radiant output). Subsequently, the herbicidal evaluation under natural sunlight was then performed. Another three groups of grass were treated as follows: (1) control group, grass received only water; (2) PQ group, grass exposed to PQ; (3) PQ-loaded vesicles group, grass exposed to PQ-loaded

vesicles, and all of these groups of grass pots were placed under natural sunlight (as shown in Supplementary Fig. 9). In all of these experiments, according to the manufacturer's instructions (Syngenta Crops, Lda), the concentration of PQ was set as 2 mg mL$^{-1}$ and the grass was sprayed with a hand sprayer (media volume 4.5 mL) to ensure a complete coverage. The spray area was limited to a plastic funnel with a diameter of 6 cm. The experiment was monitored for 5 days and the herbicidal activity was determined by visual assessment.

**Data availability**. The data that support the findings of this study are available from the corresponding author upon request.

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

## Acknowledgements

We wish to acknowledge Macau Science and Technology Development Fund (grant nos.: 069/2015/A2 and 030/2017/A1), Research Committee at the University of Macau (grant nos.: MYRG2016-00133-ICMS-QRCM, and MYRG2016-00165-ICMS-QRCM and MYRG2016-00008-ICMS-QRCM) for providing financial support to this research.

## Author contributions

C.G., S.M.Y.L. and R.W. designed the project conceptually. C.G. synthesized and characterized the vesicles, and C.G. and F.T. performed in vitro experiments. C.G., Q.H. and Y.F. performed systemic in vivo experiments. C.G., Q.L. and Y.F. performed pilot in vivo experiments. C.G., Q.H., M.P.M.H., J.Z., S.M.Y.L. and R.W. discussed and analyzed data.

C.G., S.M.Y.L. and R.W. summarized all the data and wrote the manuscript. All authors read and approved the manuscript.

## Additional information

**Competing interests:** R.W., S.M.Y.L., C.G. and M.P.M.H. are currently applying for a Chinese patent relating to the contents of this manuscript. The remaining authors declare no competing interests.

