## [Peer Review File · Nature Communications]

Reviewers' comments:

Reviewer #1 (Remarks to the Author):

Manuscript: A user-friendly herbicide derived from photo-responsive supramolecular vesicles

Authors claim that their study demonstrated, for the first time, that a photo-responsive vesicle formulation of paraquat (PQ) prepared via a facile one-pot supramolecular complexation, may provide a safe yet effective herbicide formulation, with a significant potential for applications in green agriculture. The obtained results are interesting, and with potential applicability in agricultural settings, but unfortunately flawed by some experimental and conceptual defaults:

1. In page 2, authors refer that "During the photosynthetic process in plants, PQ shuts the electron transport chain, leading to production of superoxide anions ($O_2^{\bullet-}$) which disrupt the cell membrane and kill the plants. Thus is incorrect, or at least very incomplete. PQ causes a deviation of electron flow from Photosystem I (which normally transfers its electron to ferredoxin), leading to an inhibition of oxidized nicotinamide adenine dinucleotide phosphate (NADP⁺) reduction during photosynthesis. Resulting from this process, PQ^{•+} is produced in the cell at the expense of NADPH, starting this way the prooxidant process (Dinis-Oliveira et al, 2008).
2. In page 3, authors refer that "This study provides the first safe yet effective PQ formulation". This is incorrect, as others have already developed effective PQ formulations (e.g. Dinis-Oliveira et al, 2013).
3. Another point that was not studied, but with great importance in a formulation, is its shelf stability and therefore its safety at long term. There is no point to develop a new formulation, if extended shelf life is not demonstrated.
4. From in vitro results, one can infer that PQ-loaded vesicles (without UV) are safer but still toxic at the tested concentrations or doses. It was also observed that 10% of the tested zebrafish died. One cannot consider it a safe formulation, based on these findings.
5. Survival rate in mice was only observed for 7 days. This is insufficient, as it is well known that PQ is taken up by the lungs, where it induces toxicity at long term, leading to second phase of PQ-induced lung toxicity, which involves the development of extensive fibrosis. Survival evaluation should be performed for at least 30 days.
6. Main organs studied in mice were the liver and the kidney. Lung was only addressed in a histological study, and brain was not addressed at all. This is incomprehensible, as the lung is considered the main target organ of PQ, and brain is considered of high importance in PQ research due to its potential of causing Parkinson's disease.

References:

- Baltazar MT, Dinis-Oliveira RJ, Guilhermino L, Bastos Mde L, Duarte JA, Carvalho F. New formulation of paraquat with lysine acetylsalicylate with low mammalian toxicity and effective herbicidal activity. *Pest Manag Sci.* 2013; 69(4):553-8.
- Dinis-Oliveira RJ, Duarte JA, Sánchez-Navarro A, Remião F, Bastos ML, Carvalho F. Paraquat poisonings: mechanisms of lung toxicity, clinical features, and treatment. *Crit Rev Toxicol.* 2008; 38(1):13-71.

Reviewer #2 (Remarks to the Author):

This very nice manuscript describes a photo-responsive vesicle was prepared via one-pot synthesis by the self-assembly of amphiphilic ternary supramolecular complexes between CB[8], PQ and hydrophobic trans-G. More importantly, this sunlight-responsive vesicle may provide a platform to turn highly toxic herbicides or pesticide into user-friendly formulations, with a significant potential for applications in green agriculture. I think this manuscript gives a good example for developing potential user-friendly herbicides. Therefore, I recommend it to be accepted after the authors make the following minor corrections:

1. Page 5, line 217, the authors mentioned "After incubation", but the authors did not explain how long the vesicles had a stability in the dark when they were taken up by the cells.
2. In the part of "In vivo safety evaluation in a mouse model", the authors should verify the low toxicity of PQ-loaded vesicles in vivo for a long time instead of just 7 days because if the toxicity of PQ-loaded vesicles in vivo may have the same result with the PQ for cell or mouse in the long run. In other words, the herbicide may be not real user-friendly in the long run.
3. Why did the authors not study the stability of PQ-loaded vesicles? Because the pH of the cell is not always the same?
4. An important paper about paraquat should be cited: J. Am. Chem. Soc. 2012, 134, 19489.

Reviewer #3 (Remarks to the Author):

In the manuscript by Gao et al, the authors describe the development and evaluation of a caged herbicide vesicle where paraquat can be quantitatively released by UV illumination. There are many strengths of this manuscript. First, the practical goal is to develop a technology to reduce inadvertent herbicide exposures in real world applications is laudable. This remains a substantial challenge in agriculture because many of the commonly used herbicides codes environmental and human health risks.

The manuscript is well written, logically organized and moves from the careful synthesis and characterization of the super molecular vesicles to safety assessments and performance evaluations. The preparation of the photo responsive vesicles appears to be straightforward with the production of uniform vesicles with high release performance under UV illumination. The strength of this manuscript lies in the synthesis and characterization of the vesicles.

The safety assessment experiments are limited in scope. First, the justification for using Cos-7 and L-02 cells was not well justified. The measures of ROS and MTT are quite crude measures of toxicity for in vitro platforms. More importantly, for both the cell viability and ROS measures, the "safety margin" added by encapsulation is quite minor (Figure 3), albeit statistically significant.

For in vivo assessments, the authors moved to the zebrafish assay. Specifically comparing the impact of encapsulation plus and minus UV in a larval model. The zebrafish model is now well established as a tool to assess chemical and material toxicity, thus the selection of the model is prudent. The authors use semi quantitative approaches to measure biological uptake of Cy7.5 loaded vesicles, and the data suggests a concentration response increase in the dye. The authors assume of course that the fluorescent signal is a measure of intact vesicles and not uptake of released dye. Since experiments

initiated at two days post fertilization, a time when ingestion is not a major route of uptake, it suggests that these vesicles are entering the zebrafish by epithelial uptake, rather than oral uptake. The viability data is compelling suggesting that the vesicles indeed offer substantial reduced overall toxicity, presumably by reducing paraquat exposures. Major limitation in this study is that the authors failed to measure paraquat concentrations in the zebrafish system over time and \pm UV exposure. It would be interesting to evaluate toxicity in slightly older animals where uptake by eating could more similarly mimic real world exposures of these large vesicles.

The mouse studies are limited as they use a single IP injection of the vesicles compared to PQ alone. The measures of toxicity are quite limited measuring liver lung and kidney toxicity via histopathology. The justification for the single 20 mg/kg dose is not included in the manuscript. The measure of biomarkers of effect are similarly inadequate (only TNF-alpha and IL-1Beta are measured). There are no measures of tissue dose, or any indication regarding the elimination kinetics of the vesicle compared to the free PQ. Is it possible that the toxicity could be delayed? What is the anticipated in vivo fate of the vesicles?

The weed mitigate studies do suggest that these vesicles could work in real world scenarios

Overall, this is a well-conceived set of experiments that propose a clever approach to control the release of paraquat by UV light. This approach and technology appears feasible. However, the safety assessment studies are limited, but still encouraging. The true declaration of safety of these materials will require more significant dose response set of experiments where the potential effects of these vesicles are more carefully evaluated.

Reviewer #4 (Remarks to the Author):

This is a technically sound paper that describes the construction and use of orthogonal photo-responsive supramolecular entities, for the controlled release of class two broad spectrum herbicide, Paraquat, in response to UV light, which might provide a safe yet effective herbicide formulation, with a significant potential for agricultural applications. The conclusions are clearly supported by evidence provided by a number of independent, converging results performed with high standard technical skills. Although this is, to my knowledge, the first report demonstrating the use of supramolecular vesicles for encapsulation and regulated release of an herbicide, the originality and novelty of this contribution is slightly blunted, because the development and potential therapeutic use of the orthogonal switching supramolecular complexes as drug delivery systems have already been reported (as the authors acknowledge in the appended references). The agricultural impact is relevant, as ingesting paraquat causes organ failure within several days to several weeks that can lead to death. However, diluted paraquat used for spraying is less toxic; thus the greatest risk of accidental poisoning of paraquat is during mixing and loading paraquat for use, which if explored by the authors would enhance the overall significance of their studies. Further, the novelty and the significance could also be enhanced if the authors could show the potential use of this compound in the natural setting rather than under the simulated conditions, at least with respect to the efficacy of the herbicide.

Minor comments:

1) Paraquat is used in both formation of the vesicles as well as to test the proof of principle of the concept that the system can be used to promote safe and effective herbicide formulations. In this context it would have been nice to know if supramolecular complexes formed with viologen can support other herbicides.

- 2) Panel 'h' in figure 2 requires revision. The legend does not provide sufficient information to differentiate between the inset and the main graph. It would also be nice to know the release kinetics under normal sunlight, since the simulated sunlight covers only a small part of the UV spectrum.
- 3) In figure 3 the author's suggest that the data for PQ-loaded vesicles (UV) was normalized to UV light alone values. It might be appropriate to present these results either in the main or more likely in the supplemental data figures so that the readers can evaluate the data objectively.
- 4) Although the manuscript is easy to read for the most part, it could benefit from editorial corrections to avoid occasional grammatical errors that show up in the presentation.

Response to Review

Reviewer #1

Authors claim that their study demonstrated, for the first time, that a photo-responsive vesicle formulation of paraquat (PQ) prepared via a facile one-pot supramolecular complexation, may provide a safe yet effective herbicide formulation, with a significant potential for applications in green agriculture. The obtained results are interesting, and with potential applicability in agricultural settings, but unfortunately flawed by some experimental and conceptual defaults:

1. In page 2, authors refer that “During the photosynthetic process in plants, PQ shuts the electron transport chain, leading to production of superoxide anions ($O_2^{\bullet-}$) which disrupt the cell membrane and kill the plants. Thus is incorrect, or at least very incomplete. PQ causes a deviation of electron flow from Photosystem I (which normally transfers its electron to ferredoxin), leading to an inhibition of oxidized nicotinamide adenine dinucleotide phosphate ($NADP^+$) reduction during photosynthesis. Resulting from this process, $PQ^{\bullet+}$ is produced in the cell at the expense of NADPH, starting this way the prooxidant process (Dinis-Oliveira et al, 2008).

Response: Thanks this reviewer for the important supplementation about PQ action mechanism. We revised this part of the introduction section based on the reviewer’s input and cited the relevant reference (Dinis-Oliveira et al, 2008) properly.

2. In page 3, authors refer that “This study provides the first safe yet effective PQ formulation”. This is incorrect, as others have already developed effective PQ formulations (e.g. Dinis-Oliveira et al, 2013).

Response: Thanks for the kind reminder. We agree with the reviewer that several eminent scientists such as Dinis-Oliveira et al have already developed safe, effective PQ formulations with promising potential applications. Accordingly, relevant discussion was added into Introduction section of the revised manuscript with the key reference (Dinis-Oliveira et al, 2013) cited.

In fact, the key intention of our work was to develop a unique, novel platform to turn highly toxic herbicides or pesticide into user-friendly formulations by using supramolecular photo-responsive vesicles. Thus, to avoid confusion and to reflect the actual novelty of the work, “This study provides the first safe yet effective PQ formulation” was revised to “This study provides the first safe yet effective photo-responsive vesicle formulation of PQ, and offers novel insights for user-friendly herbicides formulations in green agriculture”.

3. Another point that was not studied, but with great importance in a formulation, is its

shelf stability and therefore its safety at long term. There is no point to develop a new formulation, if extended shelf life is not demonstrated.

Response: We really appreciate the reviewer's constructive suggestion. The size stability of PQ-loaded vesicles stored for 3 and 210 days at ambient conditions (in the dark) was studied (Table S1a), and the herbicide release profile of these stored PQ-loaded vesicles exposed to natural sunshine was performed (Fig. S4a). The results showed that their size increased moderately after storage for 210 days, and more importantly, the payload release behavior under natural sunshine was sufficiently maintained, suggesting their relatively long shelf-life stability. Furthermore, these vesicles, after storage for 210 days in the dark, still preserved significant safety profile in a mouse model (updated Fig. 5 that has shown safety with 1-month follow-up after administration) and herbicidal efficacy under sunlight (Fig. S9).

4. From *in vitro* results, one can infer that PQ-loaded vesicles (without UV) are safer but still toxic at the tested concentrations or doses. It was also observed that 10% of the tested zebrafish died. One cannot consider it a safe formulation, based on these findings.

Response: Thanks for the reviewer's comment on the safety profile. Just like anything else, "safe" is a relative concept. Even with 100% survival rate, one couldn't say it is absolutely safe without full evaluation of organ toxicities and functions etc. Here in our study, "safe" formulation refers to "much safer" version in comparison with free PQ. From both *in vitro* and *in vivo* experiments, we have found that the toxicity of PQ-loaded vesicles was decreased significantly in comparison to that of free PQ. Without any compromise on the herbicidal activity, the survival rate of zebrafish incubated with this formulation increased to 90% while all zebrafish died treated with the same concentration of PQ. In the safety evaluation on mouse for 31 days (updated Fig. 5), the survival rate of mouse administrated with this formulation was 100%, in contrast to only 33% for the group treated with free PQ. The "safer" profile was supported by systemic evaluations including organs histological examinations, liver/kidney functions study, potential Parkinson disease progression, as well as examination of the levels of inflammatory factors in the lung. These results do support the new formulation is much safer in comparison to free PQ. We do not think anyone can claim their PQ formulation is "absolutely" safe across all animal models (even in the case of 100% survival rate, as 100% survival does not mean absolute safety either). Therefore, in our text, we often use the word "safer", as it is relative to free PQ, when describing the new formulation's safety profile.

5. Survival rate in mice was only observed for 7 days. This is insufficient, as it is well known that PQ is taken up by the lungs, where it induces toxicity at long term, leading to second phase of PQ-induced lung toxicity, which involves the development of extensive fibrosis. Survival evaluation should be performed for at least 30 days.

Response: Thanks very much for the constructive comment. The survival experiment in mice was repeated and evaluated for 31 days post administration, and the results were updated in the new Fig. 5.

6. Main organs studied in mice were the liver and the kidney. Lung was only addressed in a histological study, and brain was not addressed at all. This is incomprehensible, as the lung is considered the main target organ of PQ, and brain is considered of high importance in PQ research due to its potential of causing Parkinson's disease.

Response: Thanks a lot for the constructive comments. To address this comment, we have conducted further experiments on the lungs and brain (updated in Fig. 5 and Fig. S7). In addition to HE staining examination in the lung, Masson trichrome staining examination was conducted, and inflammatory factors in the lungs (such as TNF- α , IL-1 β , IL-6 and IL-10) were also determined for detailed evaluation of the lung. For the brain, we conducted histological studies on nigra and striatum. Additionally, Parkinson's disease related neurotransmitters (dopamine and dihydrophenylacetic acid) in striatum were measured by LC-MS method. Discussions about these results were added accordingly in the main text.

References:

Baltazar MT, Dinis-Oliveira RJ, Guilhermino L, Bastos Mde L, Duarte JA, Carvalho F. New formulation of paraquat with lysine acetylsalicylate with low mammalian toxicity and effective herbicidal activity. *Pest Manag Sci.* 2013; 69(4):553-8.
Dinis-Oliveira RJ, Duarte JA, Sánchez-Navarro A, Remião F, Bastos ML, Carvalho F. Paraquat poisonings: mechanisms of lung toxicity, clinical features, and treatment. *Crit Rev Toxicol.* 2008; 38(1):13-71.

Reviewer #2

This very nice manuscript describes a photo-responsive vesicle was prepared via one-pot synthesis by the self-assembly of amphiphilic ternary supramolecular complexes between CB[8], PQ and hydrophobic trans-G. More importantly, this sunlight-responsive vesicle may provide a platform to turn highly toxic herbicides or pesticide into user-friendly formulations, with a significant potential for applications in green agriculture. I think this manuscript gives a good example for developing potential user-friendly herbicides. Therefore, I recommend it to be accepted after the authors make the following minor corrections:

Response: Thanks for the reviewer's comments and recommendation for publication upon a minor revision.

1. Page 5, line 217, the authors mentioned "After incubation", but the authors did not

explain how long the vesicles had a stability in the dark when they were taken up by the cells.

Response: The stability of vesicles in the cell culture medium was studied and the result showed that the vesicles had a good stability after incubation in cell culture medium for 72 h in the dark (Table S1b). In reality, the cells were incubated in the media containing NR-loaded vesicles for only 2 h. Subsequently, the media were replaced with fresh media for additional incubation for 10, 20 and 30 min. To add clarity, “After incubation” was changed into “After incubation for 2 h”.

2. In the part of “In vivo safety evaluation in a mouse model”, the authors should verify the low toxicity of PQ-loaded vesicles in vivo for a long time instead of just 7 days because if the toxicity of PQ-loaded vesicles in vivo may have the same result with the PQ for cell or mouse in the long run. In other words, the herbicide may be not real user-friendly in the long run.

Response: Thanks for the constructive comments raised by both Reviewer 1 and 2. The *in vivo* safety evaluation of the new formulation of herbicide in mice was extended to 31 days post single-dose administration. The survival rate, body weight changes, histological studies, organ index and functional biomarkers on the liver and kidneys, inflammatory factors in the lungs, and potential PD development in the brain were all evaluated for 31 days after administration, and the results shown in updated Fig. 5. These results, together with updated discussions, have confirmed that PQ-loaded vesicles are much safer than free PQ.

3. Why did the authors not study the stability of PQ-loaded vesicles? Because the pH of the cell is not always the same?

Response: Thanks for the reminder about stability of the vesicles in different pH environments. The stability of PQ-loaded vesicles was evaluated in different medium including PBS (see the data in Table S1a), PBS with different pH values (pH 5.0-7.4) that are biologically relevant (Fig. S3), as well as cell culture medium (Table S1b). The results exhibited a good stability profile of the vesicles.

4. An important paper about paraquat should be cited: J. Am. Chem. Soc. 2012, 134, 19489.

Response: Thanks. This highly relevant paper has been cited and a brief discussion about this reference has been added in the revised manuscript.

Reviewer #3

In the manuscript by Gao et al, the authors describe the development and evaluation of

a caged herbicide vesicle where paraquat can be quantitatively released by UV illumination. There are many strengths of this manuscript. First, the practical goal is to develop a technology to reduce inadvertent herbicide exposures in real world applications is laudable. This remains a substantial challenge in agriculture because many of the commonly used herbicides codes environmental and human health risks.

The manuscript is well written, logically organized and moves from the careful synthesis and characterization of the super molecular vesicles to safety assessments and performance evaluations. The preparation of the photo responsive vesicles appears to be straightforward with the production of uniform vesicles with high release performance under UV illumination. The strength of this manuscript lies in the synthesis and characterization of the vesicles.

Response: Thanks a lot for the positive comments on the value of this paper and recognition of the strength of the work.

The safety assessment experiments are limited in scope. First, the justification for using Cos-7 and L-02 cells was not well justified. The measures of ROS and MTT are quite crude measures of toxicity for in vitro platforms. More importantly, for both the cell viability and ROS measures, the “safety margin” added by encapsulation is quite minor (Figure 3), albeit statistically significant.

Response: Thanks for the comments on cellular studies. In Results and Discussions, the justification for using COS-7 and L-02 cells is now added: “With regards to possible damage on the liver and kidney after acute PQ exposure, the toxicity of PQ-loaded vesicles on L-02 cell line (human liver cells) and COS-7 cell line (monkey kidney fibroblast cells) were firstly evaluated, and compared against that of free PQ (Toxicology, 2006, 227, 73-85)”. In order to demonstrate improved “safety margin” by the new formulation in both cell viability and ROS measurements, the concentrations of PQs (free and encapsulated) in these studied were elevated for 10-fold, and the new results have been updated in Fig. 3 showing significant improvements in toxicity by the new PQ formulation. In addition to the ROS and MTT assay, cell apoptosis experiments for both cell lines were performed, demonstrating a much improved safety profile of PQ in the new formulation, consistent with the results of MTT and ROS assays. The results have been added into the updated Fig. 3, and relevant discussions were added accordingly.

For in vivo assessments, the authors moved to the zebrafish assay. Specifically comparing the impact of encapsulation plus and minus UV in a larval model. The zebrafish model is now well established as a tool to assess chemical and material toxicity, thus the selection of the model is prudent. The authors use semi quantitative approaches to measure biological uptake of Cy7.5 loaded vesicles, and the data suggests a concentration response increase in the dye. The authors assume of course that the fluorescent signal is a measure of intact vesicles and not uptake of released dye. Since

experiments initiated at two days post fertilization, a time when ingestion is not a major route of uptake, it suggests that these vesicles are entering the zebrafish by epithelial uptake, rather than oral uptake. The viability data is compelling suggesting that the vesicles indeed offer substantial reduced overall toxicity, presumably by reducing paraquat exposures. Major limitation in this study is that the authors failed to measure paraquat concentrations in the zebrafish system over time and \pm UV exposure. It would be interesting to evaluate toxicity in slightly older animals where uptake by eating could more similarly mimic real world exposures of these large vesicles.

Response: Thanks for the valuable comments on our zebrafish assay. To address the point of “to evaluate toxicity in slightly older animals”, older zebrafish (30 dpf), known to take up food and substances orally, were recently employed to evaluate toxicity of PQ-loaded vesicles (in comparison to that of PQ). The survival rates of older zebrafish treated free PQ and PQ-loaded vesicles were studied and the data were added in Fig. 4(e). Furthermore, to address the major limitation of “the authors failed to measure paraquat concentrations in the zebrafish system”, quantitative analysis of PQ concentration in older zebrafish (30 dpf) over time with or without UV exposure were conducted, respectively by LC-MS. As shown in Table S2, the PQ concentrations in zebrafish treated with PQ-loaded vesicles at various time points (1-8 h) of incubation were generally higher than those of free PQ treated groups, likely attributed to good epithelial penetration ability of nanosized PQ, in addition to presumably equal oral uptake. In addition, there was no obvious difference in PQ uptake in zebrafish treated with PQ-loaded vesicles with or without UV irradiation, as expected (UV irradiation would only result in more free PQ in vivo, causing more serious toxicity). The survival curve of the 30-dpf zebrafish shown in Fig. 4e, indicated that almost 90% of the zebrafish treated with PQ-loaded vesicles (without UV irradiation) survived within 60 h incubation. In contrast, upon exposure of the culture media to UV light, the toxicity profile of PQ-loaded vesicles is rather similar to that of free PQ, from a survival rate perspective. Although the internalization of PQ-loaded vesicles was same between the groups with and without UV irradiation, the different mortality rates further confirmed a decent safety profile of PQ-loaded vesicles in zebrafish if not exposed to UV light.

The mouse studies are limited as they use a single IP injection of the vesicles compared to PQ alone. The measures of toxicity are quite limited measuring liver lung and kidney toxicity via histopathology. The justification for the single 20 mg/kg dose is not included in the manuscript. The measure of biomarkers of effect are similarly inadequate (only TNF-alpha and IL-1Beta are measured). There are no measures of tissue dose, or any indication regarding the elimination kinetics of the vesicle compared to the free PQ. Is it possible that the toxicity could be delayed? What is the anticipated in vivo fate of the vesicles?

Response: Thanks for the valuable comments. For the mouse studies, we further extended our study from a single dose to a series of doses as a dose-escalation study, 20 mg/kg, 30 mg/kg, 40 mg/kg and 50 mg/kg, with the survival rate in each group

followed for 10 days, for the purpose of dose evaluation and selection. These data were newly added into the ESI as Fig. S7a. With these data, we subsequently justified and decided to select 20 mg/kg dose for body-weight changes follow-up, as the majority of the mice survived in the PQ group, such that statistically meaningful body weights can be measured during 30-day study in all the groups for a fair comparison (data was updated in Fig. 5(a)). Meanwhile, we chose 40 mg/kg dose for a long-term survival study for potential delayed toxicity (>30 days), as at this dose the majority of the mice died within 4 days in the PQ group, therefore significantly improved survival rate can be, and indeed was, observed (Fig. 5(b)). We chose 30 mg/kg for systemic toxicity evaluations as these mice experienced obvious toxicity and mid-level mortality, therefore under this dose level enough organs tissue samples and blood samples can be taken and examined over a 30-day period, and compared in a statistically meaningful manner (Fig. 5 (c)-(h)).

To have more extensive and systemic toxicology evaluation of free PQ and PQ-loaded vesicles, the liver, lungs, kidneys and brain (striatum and nigra) were collected on Day 3 and Day 31, during the 31-day study, for histological examination (HE staining for all major organs and Masson staining for the lungs). These data have been summarized in Fig. 5 (g) and (h). In addition, blood samples were collected from these mice, and were analyzed for the liver function biomarkers (ALT and AST, Fig. 5 (c)) and the kidney function biomarkers (UA and BUN, Fig. 5 (d)) examinations. For the brain, we further examined the levels of Parkinson's disease related neurotransmitter and metabolite such as dopamine and dihydrophenylacetic acid in striatum by LC-MS. The data exhibited negligible toxicity in the brain by PQ in both free and vesicle formulations (Fig. 5 (e)), consistent with a previous literature report (Mol. Neurobiol. 2017, 55, 890-897) that a single acute PQ exposure wouldn't induce PD. Furthermore, we extended the analysis of inflammatory factors in the lungs to all four major cytokines including TNF- α , IL-1 β , IL-6 and IL-10 (Fig. 5 (f)).

The PQ concentrations in major organs including the liver, lungs, kidneys and brain were analyzed by LC-MS at 30 min and 7 days, respectively, after administration of PQ or PQ-loaded vesicles, with the data summarized in Table S3. The results suggested that PQ and PQ-loaded vesicles were mainly distributed in the liver and kidneys at 30 min post administration. In contrast, at Day 7 post administration, PQ was not detectable in any of the major organs, suggested that PQ and PQ-loaded vesicles were likely metabolized and excreted before Day 7. Additionally, whole-body fluorescence imaging of the mice administered with Cy7.5-loaded vesicles further confirmed that the payload was mostly all cleared out of the mice by Day 6 post dose administration (Fig. S7e).

The weed mitigate studies do suggest that these vesicles could work in real world scenarios

Overall, this is a well-conceived set of experiments that propose a clever approach to control the release of paraquat by UV light. This approach and technology appears feasible. However, the safety assessment studies are limited, but still encouraging. The

true declaration of safety of these materials will require more significant dose response set of experiments where the potential effects of these vesicles are more carefully evaluated.

Response: Thanks for the overall positive evaluation and recognition of the significant scientific values as well as the practical application of this work. As responded above already to this reviewer and others, more systemic safety assessment of PQ-loaded vesicles in both zebrafish and mouse model (particularly the mouse model with 31-day follow-up) have been conducted, and more supportive data have been added into the revised manuscript (and ESI) with expanded discussions.

Reviewer #4

This is a technically sound paper that describes the construction and use of orthogonal photo-responsive supramolecular entities, for the controlled release of class two broad spectrum herbicide, Paraquat, in response to UV light, which might provide a safe yet effective herbicide formulation, with a significant potential for agricultural applications. The conclusions are clearly supported by evidence provided by a number of independent, converging results performed with high standard technical skills. Although this is, to my knowledge, the first report demonstrating the use of supramolecular vesicles for encapsulation and regulated release of an herbicide, the originality and novelty of this contribution is slightly blunted, because the development and potential therapeutic use of the orthogonal switching supramolecular complexes as drug delivery systems have already been reported (as the authors acknowledge in the appended references). The agricultural impact is relevant, as ingesting paraquat causes organ failure within several days to several weeks that can lead to death.

Response: Thanks a lot for your kind recognition of the scientific strength and values of this work.

However, diluted paraquat used for spraying is less toxic; thus the greatest risk of accidental poisoning of paraquat is during mixing and loading paraquat for use, which if explored by the authors would enhance the overall significance of their studies. Further, the novelty and the significance could also be enhanced if the authors could show the potential use of this compound in the natural setting rather than under the simulated conditions, at least with respect to the efficacy of the herbicide.

Response: Thanks so much for your comment about possible PQ poisoning situation of using concentrated PQ. The concentration of PQ used for spraying for agricultural use is usually 2 mg/mL (which is also the concentration we used in the herbicidal study), according to the manufacturer's instruction (Syngenta Crops, Lda). In the safety evaluation on the mouse model, the concentration of PQ was in fact quite concentrated, ~ 10 mg/mL, due to the limited volume that could be administered into mice. 10 mg/mL

is rather comparable to the mixing and loading concentration of PQ that the reviewer commented. The systemic safety evaluations (both the original set of data and the newly extended data to address all of the reviewers' comments) exhibited a much higher safety profile from PQ-loaded vesicles, in contrast to free PQ.

In order to demonstrate that the novel herbicide formulation also works under a natural setting, with respect to the efficacy of the herbicide. The herbicide release behavior of PQ-loaded vesicles under natural sunlight was recently determined (Fig. S4a). Of an interesting note, it required only 4 h to reach a cumulative release of 80% PQ under natural sunlight for PQ-loaded vesicles, which was much shorter than that of simulated sunlight. This expedited release under natural sunlight is likely attributed to the wide spectral window (290-3200 nm) of natural sunlight in comparison to our simulated sunlight (360-800 nm). Furthermore, the herbicidal efficacy of PQ-loaded vesicle under natural sunlight was evaluated and the results suggested that the novel formulation has nearly identical herbicidal efficacy with free PQ (Fig. S9), confirming that these photo-responsive vesicles may provide a promising and feasible/practical platform for sunlight-sensitive herbicide release.

Minor comments:

1) Paraquat is used in both formation of the vesicles as well as to test the proof of principle of the concept that the system can be used to promote safe and effective herbicide formulations. In this context it would have been nice to know if supramolecular complexes formed with viologen can support other herbicides.

Response: Thanks a lot for this constructive suggestion. In principle, the sunlight sensitive supramolecular complexes formed with viologen could be extended to other herbicides. To demonstrate the proof of principle of the versatility of the platform, diquat, a nonselective, fast-acting desiccant herbicide, was encapsulated in the supramolecular vesicles (referred as to “diquat-loaded vesicles”). The herbicide (diquat) release profile under natural sunlight was evaluated, and the result (Fig. S4b) was comparable to that of PQ-loaded vesicles upon exposure to sunlight (Fig. S4a). Therefore, the reviewer is right that these sunlight-responsive vesicles may provide a unique, novel platform to turn highly toxic herbicides into user-friendly formulations that have been highly desired for decades in green agriculture.

2) Panel ‘h’ in figure 2 requires revision. The legend does not provide sufficient information to differentiate between the inset and the main graph. It would also be nice to know the release kinetics under normal sunlight, since the simulated sunlight covers only a small part of the UV spectrum.

Response: Thanks for the constructive suggestion. Fig. 2h was improved per the request. The release kinetics of the herbicides-loaded supramolecular vesicles under natural sunlight were evaluated for both paraquat and diquat as model herbicides (Fig. S4a and Fig. S4b).

3) In figure 3 the author's suggest that the data for PQ-loaded vesicles (UV) was normalized to UV light alone values. It might be appropriate to present these results either in the main or more likely in the supplemental data figures so that the readers can

evaluate the data objectively.

Response: Thanks. To add clarity, the data of cell viability for UV light alone has been added into Fig. 3a now.

4) Although the manuscript is easy to read for the most part, it could benefit from editorial corrections to avoid occasional grammatical errors that show up in the presentation.

Response: Thanks for the comments. We have carefully gone over the revised manuscript several rounds to further improve the overall quality of the writing and we have also asked a native English speaker (Canadian) to help proofread and cross-check the scientific writing for now. .

REVIEWERS' COMMENTS:

Reviewer #1 (Remarks to the Author):

Authors have adequately addressed most of concerns raised by reviewers. There is only a point that requires due attention before my final advice on manuscript acceptance:

- Most of paraquat related deaths occur after ingestion of the pesticide. In the animal studies, authors used an i.p. administration of the formulation, which does not correspond to the real scenario situation and therefore cannot be used as the proof of concept to confirm the safety of the formulation. In vivo studies in mice should follow the oral route of administration.

Reviewer #2 (Remarks to the Author):

This revised manuscript can be published as it is now.

Reviewer #3 (Remarks to the Author):

The safety assessments completed in this revised manuscript are much more compelling and the manuscript has the potential to be impactful to the field.

Reviewer #4 (Remarks to the Author):

The author's have addressed the comments and concerns raised by this reviewer.

Response Letter To Review

Reviewer #1

Authors have adequately addressed most of concerns raised by reviewers. There is only a point that requires due attention before my final advice on manuscript acceptance:

- Most of paraquat related deaths occur after ingestion of the pesticide. In the animal studies, authors used an i.p. administration of the formulation, which does not correspond to the real scenario situation and therefore cannot be used as the proof of concept to confirm the safety of the formulation. In vivo studies in mice should follow the oral route of administration.

Response: We agree that most of the acute PQ poisonings take place via oral ingestion; however, chronic PQ poisonings are often via respiratory and dermal exposures. With regard to toxicity evaluations of PQ or PQ formulations, a variety of administration routes have been employed in mice or rats, including oral, subcutaneous, dermal, inhalation, and intraperitoneal routes (*Toxicol. Appl. Pharmacol.* **33**, 450-460 (1975), *Environ. Health Persp.* **15**, 1448-1453 (2007), and *Intensive Care Med.* **26**, 981-987 (2000)). Due to fast absorption into the systemic circulation and high PQ bioavailability, administration of PQ via intraperitoneal route (*Neuro Toxicol.* **37**, 1-14 (2013)) is a worse-case scenario of PQ poisoning, in comparison with that via oral route. Thus, safety evaluation in mice via intraperitoneal administration still support the superior safety profile of PQ-loaded vesicles in comparison with that of PQ in our head-to-head comparative studies. To address this comment in the main text, we have inserted a caveat that the PQ administration route in our mice study is intraperitoneal, and added brief discussions about various administration routes previously employed for toxicity evaluations of PQ, in the section of “in vivo safety evaluation in a mouse model”.

Reviewer #2

This revised manuscript can be published as it is now.

Reviewer #3

The safety assessments completed in this revised manuscript are much more compelling and the manuscript has the potential to be impactful to the field.

Reviewer #4

The author's have addressed the comments and concerns raised by this reviewer.